# Evaluation of Serum Diamine Oxidase as a Diagnostic Test for Histamine Intolerance

**DOI:** 10.3390/nu15194246

**Published:** 2023-10-02

**Authors:** Kristina Arih, Nina Đorđević, Mitja Košnik, Matija Rijavec

**Affiliations:** 1Faculty of Medicine, University of Ljubljana, 1000 Ljubljana, Slovenia; 2Laboratory for Clinical Immunology and Molecular Genetics, University Clinic of Respiratory and Allergic Diseases Golnik, 4204 Golnik, Slovenia; 3Biotechnical Faculty, University of Ljubljana, 1000 Ljubljana, Slovenia

**Keywords:** histamine intolerance, diamine oxidase, serum, diagnosis

## Abstract

Histamine intolerance (HIT) is a clinical condition caused by decreased intestinal degradation of ingested histamine, primarily due to reduced enzyme diamine oxidase (DAO) activity, leading to histamine accumulation and causing various clinical manifestations. The measurement of serum DAO is commonly used as the main diagnostic test for HIT, although its diagnostic use is still uncertain. In this retrospective study, we aimed to assess the validity of DAO determination in patients with clinically suspected HIT. We measured DAO levels in 249 patients with suspected HIT and 50 healthy adult controls without HIT-related problems. Based on five clinical criteria, we divided patients into two groups: high (all five inclusion criteria; 41 patients) and low probability of HIT (≤4 inclusion criteria; 208 patients). Patients with a “high probability of HIT” had the lowest DAO (median: 8 U/mL, IQR: 6–10) in comparison to patients with a “low probability of HIT (median: 10 U/mL, IQR: 7–16, *p* = 0.0006) and healthy controls (median: 18 U/mL, IQR: 14–22, *p* < 0.0001). The specificity and sensitivity for DAO levels < 3/< 10 U/mL (manufacturer’s set cut-off) to discriminate between patients with ‘‘high probability of HIT’’ and healthy controls were 100%/92% and 2%/71%. On the other hand, the specificity and sensitivity to discriminate between patients with ‘‘high probability of HIT’’ and ‘‘low probability of HIT’’ were 97%/61% and 2%/71%, respectively. Serum DAO determination represents an additional asset to the diagnosis of HIT based on clinical evaluation and assessment, but the diagnosis should not solely rely on DAO measurements.

## 1. Introduction

Histamine intolerance (HIT) is caused by decreased intestinal inactivation and degradation of ingested histamine. Histamine is a biogenic amine widely present in the human organism from endogenous and exogenous sources [1]. It is metabolised by two main enzymes: histamine-N-methyl transferase (HNMT) and diamine oxidase (DAO). HNMT is a cytosolic enzyme whose role is the degradation of endogenous histamine [2]. The main enzyme for the degradation of exogenous histamine is DAO, mainly expressed in epithelial cells of the small intestine, the placenta, the kidneys, and the liver [1,3,4,5]. It is a secretory enzyme stored in vesicular structures within epithelial cells and secreted into the bloodstream after a stimulation signal [3]. In normal conditions, DAO is present in the bloodstream in low concentrations; its basal concentration correlates with the level of intestinal integrity. Normal serum DAO concentrations are between 15 and 50 U/mL [6,7]. DAO is not involved only in the degradation of histamine but also in the metabolism of other biogenic amines, for which it has an even greater affinity [4].

Other biogenic amines present in food, such as monoamine tyramine, diamines putrescine and cadaverine, as well as polyamines spermine and spermidine, can affect DAO activity and consequently cause HIT symptoms. Biogenic amines contribute to histamine toxicity due to the saturation of degradation enzymes in the intestinal epithelium; therefore, the proposed diet for HIT consists of foods low in all biogenic amines, not just histamine [8,9].

Inflammatory intestine diseases, such as inflammatory bowel diseases, lactose intolerance, and celiac disease, can also impair DAO activity. Inflammation causes damage to the intestinal mucosa, leading to decreased DAO expression and activity, which can lead to secondary HIT. The severity of epithelium damage correlates with the level of diminished DAO activity [10]. Secondary HIT can also be caused by different medications that disrupt DAO activity, such as antibiotics, antimalarials, antituberculotics, H2 receptor antagonists, antihypertensives, analgesics, mucolytics, antidepressants, antiemetics, and muscle relaxants [1,11].

Decreased DAO activity leads to histamine accumulation, making HIT symptoms and signs appear. Various organic systems are affected due to the ubiquitous distribution of the four histamine receptors in different tissues and organs. We can divide symptoms into six groups: gastrointestinal (diarrhoea, constipation, vomiting, and abdominal pain), cardiovascular (hypotension and arrhythmias—tachycardia), cutaneous (pruritus, urticaria, flushing), respiratory (cough, bronchospasm, rhinitis, and sinusitis), ocular (conjunctivitis), and others (headache, heat waves, swollen joints, oral ulcers, and hand paraesthesia) [1,2,12]. Symptoms appear between 2 h and one day after the consumption of foods rich in biogenic amines [8]. The most frequent foodstuffs rich in biogenic amines are fermented food (sauerkraut), chocolate, alcoholic drinks (red wine and champagne), cheese (aged cheese), meat (cured meat), conserved food (especially fish), vegetables (tomato, aubergine, and spinach), fruits (pineapple, grapefruit, and kiwi), and nuts [12,13]. The concentration of histamine depends on the food preparation process and storage. Bacteria involved in lactic acid fermentation and food spoilage produce additional histamine. Impaired bacterial activity due to added NaCl results in a lower concentration of histamine. Lower pH, higher temperature, and prolonged food exposure to bacteria increase histamine concentration [2]. We can use bacteria that are not producing biogenic amines to prevent histamine accumulation in food fermentation. Another alternative, which is still in the research process, is adding a microorganism that expresses the enzyme DAO to degrade accumulated biogenic amines in food [1,2,14].

The diagnostic work-up of HIT is complex and challenging due to broad clinical manifestations involving multiple organs and a lack of information about in vitro and in vivo diagnostic tests for HIT. Currently, the diagnosis is mainly achieved clinically, consisting of, at the minimum, two typical symptoms: improvement of symptoms while following a low-biogenic amine diet and treatment with antihistamines [12]. The most studied and frequently used diagnostic test for HIT is the determination of serum DAO concentration and activity. Still, there are some doubts about whether the test is suitable for diagnosis. In addition to DAO determination, other tests were proposed for HIT diagnosis, such as the histamine 50-prick test, an intestinal biopsy, the histamine provocation test, or the histamine metabolomics in urine. Still, data about their usefulness as diagnostic tests is lacking [1]. Due to challenges in correctly recognising the disorder, the prevalence of HIT is frequently underestimated, predicted to be between 1 and 3%. The majority of patients (80%) with this condition are middle-aged [1,8,12]. The most efficient measure to control HIT symptoms is following a low-biogenic amine diet, which is beneficial for improving symptoms and increasing serum DAO levels and activity [15]. Exogenous DAO supplementation and treatment with antihistamines might also improve HIT symptoms. With improved serum DAO activity and a lower histamine concentration, patients can be less strict with a low-biogenic amine diet [14].

This retrospective study aimed to assess the validity of DAO determination in patients with clinically suspected HIT. We measured serum DAO levels in patients with suspected HIT with different degrees of symptoms and in healthy control subjects with no HIT-related problems. We assessed the optimal threshold and reference range that were most suitable to distinguish actual patients from controls.

## 2. Materials and Methods

### 2.1. Patients

A total of 249 patients and 50 healthy adult controls were included in the final analysis of this retrospective study. We enrolled 300 consecutive adult patients with suspected HIT evaluated at the University Clinic of Respiratory and Allergic Diseases Golnik. DAO levels were measured in serum between November 2017 and December 2020. Based on clinical data collected from the hospital information system, we classified patients into distinct groups: ‘‘high probability of HIT’’ (41 patients) and ‘‘low probability of HIT’’ (208 patients). Twenty-one patients were excluded from the research due to insufficient clinical data and 30 due to secondary HIT (lactose intolerance/celiac disease). We enrolled 50 healthy adults without any HIT-related problems who were age- and sex-matched with the included patients. In the healthy adult control group, subjects with a history of food intolerances, IgE-mediated food allergies, celiac disease, and gastric acid hypersecretory states were also excluded. The study was conducted according to the Declaration of Helsinki. It was approved by the Slovenian National Medical Ethics Committee (approval number 0120-155/2021/3), and all patients gave their informed written consent.

We classified patients based on five inclusion criteria: typical clinical manifestations (gastrointestinal, cardiovascular, cutaneous, respiratory, ocular, etc.), the appearance of symptoms after consumption of biogenic amine-rich food, the appearance of symptoms within 2 h to 1 day, improvement of symptoms while following a low-biogenic amine diet, and improvement of symptoms through treatment (with antihistamines or exogenous DAO supplementation). Patients who fulfilled all inclusion criteria were classified in the ‘‘high probability of HIT’’ group, and others who fulfilled 4 or fewer were classified in the ‘‘low probability of HIT’’ group.

### 2.2. Determination of Serum DAO Concentration

The serum DAO concentration was determined using a quantitative Enzyme-Linked ImmunoSorbent Assay ELISA and a set of reagents IDK DAO ELISA (Immunodiagnostik AG, Germany) in accordance with the manufacturer’s instructions. The reference ranges defined by the assay manufacturer were: <3 U/mL: high incidence for HIT; 3–10 U/mL: HIT probable; >10 U/mL: low HIT probability.

### 2.3. Statistical Analysis

Statistical analysis was performed using IBM SPSS Statistics software version 25 (SPSS Inc., Chicago, IL, USA) and GraphPad Prism 9 (GraphPad Software, Boston, MA, USA). The distribution of DAO concentration was assessed using the Kolmogorov-Smirnov test of normality. Numeric data were presented with medians and interquartile ranges (IQR). Statistical significance between different groups was determined with the Mann-Whitney, Kruskal-Wallis, and Chi-square tests. Possible correlations between variables were determined with Pearson’s chi-squared test. *p*-values below 0.05 were considered significant.

We evaluated the performance of the IDK DAO ELISA (Immunodiagnostik AG, Bensheim, Germany) test for the diagnosis of HIT using receiver operating characteristic (ROC) curve analysis.

## 3. Results

### 3.1. Characteristics of the Study Population and Serum DAO Levels

Among the 249 patients with suspected HIT (73% female; median age 47 years (IQR 38–61)) included in the final analysis, 41 (16%) were classified as “high probability of HIT” (81% female, median age 51 years (IQR 42–66)) and 208 (84%) as “low probability of HIT” (71% female, median age 47 years (IQR 37–60)) (Table 1). There were no differences in age and sex distribution between different groups (Table 1).

In the “low probability of HIT” group, 59 patients did not fulfil any of the inclusion HIT criteria; 66 fulfilled only one; 57 fulfilled two; 25 fulfilled three; and one patient fulfilled four inclusion criteria. Among the inclusion criteria, typical clinical manifestations were the most common, present in 118 patients (Table 2, Figure 1).

### 3.2. Serum Level of DAO

Patients with a “high probability of HIT” have the lowest DAO (median: 8 U/mL, IQR: 6–10) in comparison with patients with a “low probability of HIT” (median: 10 U/mL, IQR: 7–16, *p* = 0.0006) and healthy controls (median: 18 U/mL, IQR: 14–22, *p* < 0.0001). Interestingly, patients with a “low probability of HIT” have lower DAO levels than healthy controls (*p* < 0.0001) (Figure 2). Additionally, among the patients with a “low probability of HIT”, we observed a weak negative correlation between DAO concentration and the number of inclusion clinical criteria fulfilled (*p* = 0.021, r = −0.160), as patients fulfilling three or four inclusion criteria have lower DAO than patients fulfilling two or fewer inclusion criteria (median: 10 U/mL, IQR: 5–10, vs. median: 11 U/mL, IQR: 8–17; *p* = 0.034).

### 3.3. Validity of DAO Determination for Diagnosis of HIT

Using the data from the healthy control groups, the determined specificity of the test for the two different reference ranges defined by the assay manufacturer was 100% (<3 U/mL) and 92% (<10 U/mL). Using the same reference ranges, we determined test sensitivity from the patients’ data with a ‘‘high probability of HIT’’. Sensitivity was 2% (<3 U/mL) and 71% (<10 U/mL). Hence, the reference range proposed by the manufacturer as high incidence for HIT (<3 U/mL) has good specificity (100%) but poor sensitivity (2%) indeed. While the proposed reference for HIT probable (<10 U/mL), suggesting that HIT is likely to be present, has much better sensitivity (71%) with still reasonable specificity (92%).

On the other hand, the calculated sensitivity and specificity to discriminate between patients with ‘‘high probability of HIT’’ and ‘‘low probability of HIT’’ were 71% and 61% for DAO levels <10 U/mL and 2% and 97% for DAO levels <3 U/mL, respectively.

Similarly, the sensitivity and specificity to discriminate between patients with a “high probability of HIT” and a combined “low probability of HIT plus healthy controls” were limited. Sensitivity was 71% and specificity was 67% for DAO levels <10 U/mL and 2% and 97% for <3 U/mL, respectively. 

As indicated by the estimated area under the ROC curve (AUC), DAO measurements were the most accurate in discriminating between patients ‘‘high probability of HIT’’ and healthy controls (AUC 0.872 (95% CI, 0.79–0.95), sensitivity 93% (95% CI, 81–97%), and specificity 76% (95% CI, 63–86%); cut-off 13.5 U/mL). On the other hand, the ability to discriminate between patients with “high and low probability of HIT” was limited (AUC 0.667 (95% CI, 0.58–0.75), sensitivity 71% (95% CI, 56–82%), and specificity 61% (95% CI, 54–67%); cut-off 9.5 U/mL) (Figure 3).

## 4. Discussion

We conducted a retrospective study, including patients with suspected HIT and healthy controls without any HIT-related problems. We aimed to assess the validity of DAO determination in patients with clinically suspected HIT. Based on clinical criteria, we divided patients into two groups: low (≤ 4 inclusion criteria) and high probability of HIT (all five inclusion criteria, specifically typical clinical manifestation, the appearance of symptoms after consumption of biogenic amine-rich food, the appearance of symptoms within 2 h to 1 day, improvement of symptoms while following a low-biogenic amine diet, and improvement of symptoms through treatment). We found that patients with a “high probability of HIT” have the lowest DAO concentrations compared to patients with a “low probability of HIT” and healthy controls. Most patients with a “high probability of HIT” had DAO concentrations below 10 U/mL, similar to those reported previously [16]. DAO measurements were the most accurate in discriminating between patients with ‘‘high probability of HIT’’ and healthy controls with high specificity using the reference range defined by the assay manufacturer. In contrast, the ability to discriminate between patients with “high probability of HIT” and “low probability of HIT” was limited.

Histamine intolerance is a condition that causes a broad spectrum of symptoms and signs, ranging from gastrointestinal to cutaneous, respiratory, cardiovascular, ocular, and others. These various clinical manifestations make the diagnosis of HIT peculiarly challenging. [1] The diagnosis remains clinical and would be much more effortless if a reliable diagnostic test for HIT existed. The most studied and often used clinical diagnostic test is the determination of serum DAO levels and/or activity. However, some doubts remain about whether the test is suitable for diagnosis [1,12,17].

The diagnosis of HIT has been challenging and remains so. With the clinical determination of diagnosis, we face an obstacle with the possibility of confusing HIT with other similar conditions such as various food intolerances, irritable bowel disease, and non-celiac gluten sensitivity. Frequently, those conditions are combined with HIT, which is the most likely condition secondary to this primary underlying condition. There is also limited knowledge about low DAO levels in healthy adults who do not present any HIT-related symptoms or signs [1,12]. 

In addition, it is important to state the limitations of this study. Only one DAO level measurement per patient was performed during the study. We conclude that the latter may present a limitation due to the potential overlay between distinct patient populations regarding food intake and following elimination diets. Secondly, there is a lack of knowledge of how DAO serum concentration correlates with DAO intestinal concentration and if the correlation is even significant. 

Collectively, our findings suggest that the determination of DAO serum activity can be a useful adjunct to clinical diagnosis. In patients who fulfilled all clinical criteria for HIT, DAO levels below the normal range provide evidence for HIT; however, it does not definitively confirm the diagnosis of HIT. Extremely low DAO levels (below 3 U/mL) would suggest that a patient has HIT.

Our study has significantly contributed to understanding histamine intolerance and the challenges involved in its diagnosis. We have found that measuring serum DAO levels effectively differentiates patients with a high probability of HIT from healthy controls. However, it has limited ability to distinguish between patients with a high probability and the largest group of patients, those with a low probability of HIT. Hence, the use of DAO determination alone without appropriate clinical evaluation is not suitable for the diagnosis of HIT. It could be valuable only as an additional helpful asset to the clinically based diagnosis. It is important to emphasise that, currently, the diagnosis of HIT still relies primarily on clinical evaluation. In contrast to our study, recent research found no association between DAO levels and reported HIT symptoms in histamine-rich foods [17]. This research suggests that DAO level measurement is not appropriate for diagnosing HIT. However, we assert that it might hold value as a supplementary and supportive tool alongside clinically grounded diagnoses.

Similar contradictory results on the usefulness of DAO measurements in diagnosing HIT have been reported previously since certain studies: Töndury et al. (2008) [18], Kofler et al. (2009) [19], and Schnoor et al. (2013) [20] concluded that there was no significant association between the clinical history of patients displaying typical symptoms of histamine intolerance and blood DAO activity values. These studies suggest that until further research validates its effectiveness, this technique as a routine diagnostic tool cannot be recommended. Conversely, the other three studies, Manzotti et al. (2016) [6], Mušič et al. (2013) [12], and Pinzer et al. (2018) [21], confirmed the usefulness of measuring serum DAO activity in identifying individuals who exhibit symptoms related to histamine intolerance. Furthermore, previous studies demonstrated that in patients suspected of having HIT, decreased DAO activities were correlated with elevated histamine [21], and a histamine-free diet led to histamine intolerance symptom amelioration and an increase in DAO activity [12]. Moreover, in the study by Manzotti et al. (2016) [6], serum DAO levels were found to correlate inversely with symptom severity, as symptom severity and frequency were higher in patients with lower DAO [6], further supporting the usefulness of serum DAO measurement in HIT diagnosis.

## 5. Conclusions

The results obtained from this analysis contribute to understanding the potential utility of DAO determination as a diagnostic tool for HIT. Our findings indicate that measuring serum DAO is indeed effective in distinguishing between patients with a high probability of HIT and healthy individuals. However, its capability to discriminate between patients with a high probability of HIT and those with a low probability of HIT is limited. Consequently, relying solely on DAO determination without appropriate clinical evaluation is not suitable for diagnosing HIT. Nonetheless, DAO determination can serve as a valuable additional tool to support clinically based diagnosis. It is essential to emphasise that the current diagnosis of HIT primarily relies on clinical evaluation, and the integration of DAO determination with clinical assessment can offer a more comprehensive approach to diagnosing HIT.

## Figures and Tables

**Figure 1 nutrients-15-04246-f001:**
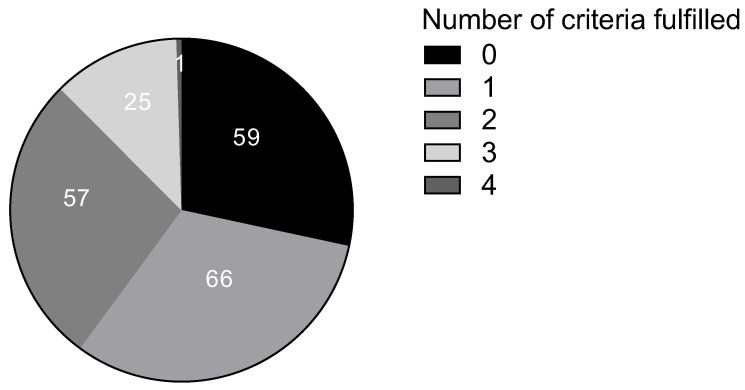
Number of “low probability of HIT” patients based on the number of inclusion criteria fulfilled.

**Figure 2 nutrients-15-04246-f002:**
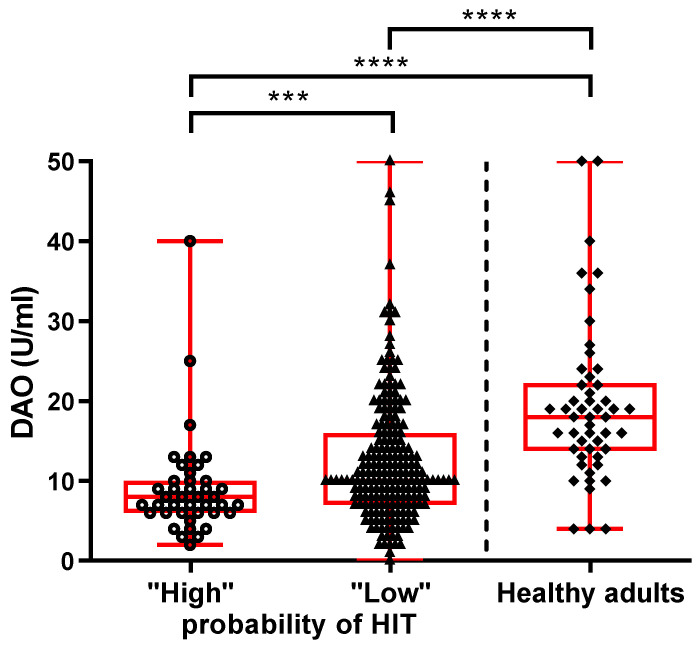
Serum DAO levels in patients with suspected HIT and healthy controls. The data are presented as medians with an interquartile range. Mann-Whitney; *** *p* < 0.001, **** *p* < 0.0001.

**Figure 3 nutrients-15-04246-f003:**
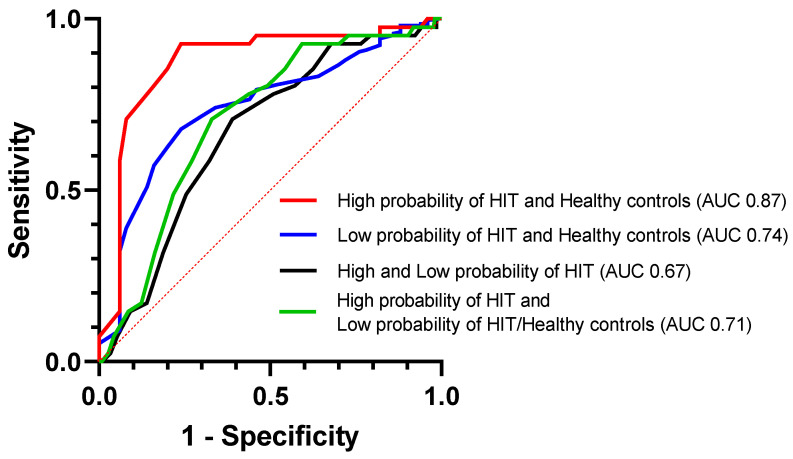
ROC curves comparing the discriminating performance of DAO.

**Table 1 nutrients-15-04246-t001:** Characteristics of the study group.

	All Patients (N = 249)	‘‘High Probability of HIT’’ (N = 41)	‘‘Low Probability of HIT’’ (N = 208)	Healthy Adults (N = 50)	*p* Value
**Age (years):** median (IQR)	47 (38–61)	51 (42–66)	47 (37–60)	44 (36–55)	0.427 ^a^
**Women: N (%)**	181 (73)	33 (81)	148 (71)	37 (74)	0.672 ^b^
**Men: N (%)**	68 (27)	8 (19)	60 (29)	13 (26)	
**DAO (U/mL): median (IQR)**	10 (7–14)	8 (6–10)	10 (7– 16)	18 (14–22)	<0.0001 ^a^

HIT, histamine intolerance; IQR, interquartile range; ^a^ Kruskal-Wallis test; ^b^ Chi-square test.

**Table 2 nutrients-15-04246-t002:** “Low probability of HIT” patients fulfilling different inclusion criteria.

Criterion	Number (%) of “Low Probability Patients”
Typical clinical manifestation	118 (57)
The appearance of symptoms after consumption of biogenic amine-rich food	30 (14)
The appearance of symptoms within 2 h to 1 day	44 (21)
Improvement of symptoms while following a low-biogenic amine diet	1 (0.5)
Improvement of symptoms through treatment (with antihistamines or exogenous DAO supplementation)	64 (31)

## Data Availability

The data supporting the findings of this study are available within the manuscript. Any additional data are available from the corresponding author upon reasonable request.

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
