# Peer review of "Evaluation of Serum Diamine Oxidase as a Diagnostic Test for Histamine Intolerance"

_nutrients, 2023, doi:10.3390/nu15194246_

Round 1

Reviewer 1 Report

In this manuscript Kristina Arih and co-authors assessed the validity of DAO determination in a cohort of patients with clinically suspected HIT

The topic is a relevant contribution in the field, where several issues are  often debated by specialists (mainly allergists and gastroenterologists). The big questions here is related to the fact that while there is a strong rational foundation around this condition, scarse and often contradictory evidence-based data are available

The actual possibility to reduce hetherogeneous phenotypes of histamine intolerance in different patients to the same pathogenic condition is challenging. The question can be put in reductionistic terms by saying that in the only study (to my knowledge) where a double blind, placebo controlled, cross-over histamine challenge was performed in vivo in healthy volunteers, ingestion of 75 mg of liquid histamine failed to reproduce histamine-associated symptoms in many subjects (Komericki P et al https://pubmed.ncbi.nlm.nih.gov/21165702/)

In the present retrospective study, inclusion criteria properly took into account the heterogeneity of HIT by considering symptoms spanning throughout 5 broad domains. This is reflecting clinical practice, where actually most specialists do agree about the convergence in single patients of so many clues to HIT that a role of biogenic amines in their clinical condition seems indisputable

The authors measure the performance of one ELISA-based, serum DAO determination assay in this conceptual framework, which currently appears to be the most prudent and at the same time the most convincing

The manuscript is well written, the results are interesting, the end-points are properly established, the discussion is elaborated with proper consideration of available literature, the statistical analysis is carried out properly, the conclusions are convincing

Author Response

Response to Reviewer 1 Comments

Comments and Suggestions for Authors

In this manuscript Kristina Arih and co-authors assessed the validity of DAO determination in a cohort of patients with clinically suspected HIT

The topic is a relevant contribution in the field, where several issues are  often debated by specialists (mainly allergists and gastroenterologists). The big questions here is related to the fact that while there is a strong rational foundation around this condition, scarse and often contradictory evidence-based data are available

The actual possibility to reduce hetherogeneous phenotypes of histamine intolerance in different patients to the same pathogenic condition is challenging. The question can be put in reductionistic terms by saying that in the only study (to my knowledge) where a double blind, placebo controlled, cross-over histamine challenge was performed in vivo in healthy volunteers, ingestion of 75 mg of liquid histamine failed to reproduce histamine-associated symptoms in many subjects (Komericki P et al https://pubmed.ncbi.nlm.nih.gov/21165702/)

In the present retrospective study, inclusion criteria properly took into account the heterogeneity of HIT by considering symptoms spanning throughout 5 broad domains. This is reflecting clinical practice, where actually most specialists do agree about the convergence in single patients of so many clues to HIT that a role of biogenic amines in their clinical condition seems indisputable

The authors measure the performance of one ELISA-based, serum DAO determination assay in this conceptual framework, which currently appears to be the most prudent and at the same time the most convincing

The manuscript is well written, the results are interesting, the end-points are properly established, the discussion is elaborated with proper consideration of available literature, the statistical analysis is carried out properly, the conclusions are convincing

Response: Thank you for carefully examining and appreciating our manuscript, which showed that serum DAO determination represents an additional asset to the diagnosis of HIT.

Reviewer 2 Report

In this work, the activity of diamine oxidase (DAO) was determined to reconfirm its clinical significance as the primary diagnostic test for histamine intolerance (HIT). This study aimed to assess the validity of DAO determination in patients with clinically suspected HIT. The authors measured serum DAO levels in patients with suspected HIT with different degrees of symptoms and in healthy control subjects with no HIT-related problems, and then assessed the optimal threshold and the reference range that was the most suitable to discriminate actual patients from controls.

 The experiments were designed and carried out without any critical errors, and English seemed plain and easy to understand. Although this study was not exactly novel, but it would be possible, or even reasonable to consider that the results obtained from this study might be able to contribute to the clinical diagnosis of HIT, thereby being worth publishing.

 The manuscript seemed almost fine except several unnatural and inappropriate points, and it would probably be better to polish it a little more.

 “Abstract”

The aim, plan and inprementation of the experiments were concisely written, and the obtained results and their interpretation were properly explained. Therefore, the abstract was easy to understand, and could be considered to be well written,

 “1. Introduction”

Overall, this section was considered to satisfactorily provide the background and purpose of this study. Particularly, it would be clear what the authors focused on, and therefore it could be easily understand the aim of this study. However, there were several points necessary to reconsider the revision described below.

1) The phrase “intestine mucosa” (line 54) might correctly be “intestinal mucosa”.

2) The phrase “causing HIT symptoms and signs to appear” (lines 60-61) would probably be possible to say “making HIT symptoms and signs appear”.

3) The phrase “Most frequent food” (line 68) would probably be better to say “Most frequent foodstuff” by guessing its meaning of sentence.

4) The meaning of the phrase “the optimal cut-offs and reference range that best discriminated” (lines 89-90) seemed easily and clearly understandable, but it seemed a little unnatural and weird, especially its wording. This phrase would probably be better to revise like “the optimal threshold and reference range that was the most suitable to discriminate actual patients from controls”.

 “2. Materials and Methods”

 This section seemed to have no critical mistakes and defects, and the assay method was not new. So there was nothing to say specifically.

 “3. Results”

The obtained results were presented here in a good manner.

1) The phrase “in comparison to” (line 157) would probably say “in comparison with” or “as compared to”.

2) The word “probable” (line 178) was suspected to be “probability”.

 “4. Discussion”

Some parts of the Discussion seemed just repetition and the same to the description in other sections. To make the manuscript more concise and compact, it would probably be better, might be necessary to avoid such a repetition or try to rephrase those parts. The descriptive contents of this section seemed quite careful and well-considered. Particularly,

1) The phrase “most likely” (lines 220-221) might be better to say “the most likely condition” to make the meaning clear and easier to understand.

2) The phrase “evidence in favour of” (line 232) seemed a little unnatural and awkward, and it would probably be better to say simply “evidence for”.

3) The phrase “the diagnosis as HIT” (line 233) should be “the diagnosis of HIT”.

4) The phrase “the utility of measuring serum DAO activity in identifying” (line 255) seemed awkward, and it would probably be better to say “the usefulness of measuring serum DAO activity to identify”.

 “5. Conclusion”

The conclusion derived from the present study might be understandable and reasonable, and therefore publishable in this journal.

The manuscript seemed almost fine except several unnatural and inappropriate points, and it would probably be better to polish it a little more.

Author Response

Response to Reviewer 2 Comments

Comments and Suggestions for Authors

In this work, the activity of diamine oxidase (DAO) was determined to reconfirm its clinical significance as the primary diagnostic test for histamine intolerance (HIT). This study aimed to assess the validity of DAO determination in patients with clinically suspected HIT. The authors measured serum DAO levels in patients with suspected HIT with different degrees of symptoms and in healthy control subjects with no HIT-related problems, and then assessed the optimal threshold and the reference range that was the most suitable to discriminate actual patients from controls.

 The experiments were designed and carried out without any critical errors, and English seemed plain and easy to understand. Although this study was not exactly novel, but it would be possible, or even reasonable to consider that the results obtained from this study might be able to contribute to the clinical diagnosis of HIT, thereby being worth publishing.

 The manuscript seemed almost fine except several unnatural and inappropriate points, and it would probably be better to polish it a little more.

 “Abstract”

The aim, plan and inprementation of the experiments were concisely written, and the obtained results and their interpretation were properly explained. Therefore, the abstract was easy to understand, and could be considered to be well written,

Response: Thank you for your careful examination and valuable suggestions for improving our manuscript. We have corrected and/or rephrased the text based on your suggestion. Please find the detailed responses below and the corresponding revisions/corrections highlighted/in track changes in the re-submitted files.

We have also added additional paragraphs in the Introduction and Discussion sections based on Editorial office recommendations.

 “1. Introduction”

Overall, this section was considered to satisfactorily provide the background and purpose of this study. Particularly, it would be clear what the authors focused on, and therefore it could be easily understand the aim of this study. However, there were several points necessary to reconsider the revision described below.

Comments 1: The phrase “intestine mucosa” (line 54) might correctly be “intestinal mucosa”.

Response 1: Thank you for pointing this out. We have corrected to intestinal mucosa (line 54 of the revised manuscript).

Comments 2: The phrase “causing HIT symptoms and signs to appear” (lines 60-61) would probably be possible to say “making HIT symptoms and signs appear”.

Response 2: We have corrected the sentence to “making HIT symptoms and signs appear” (lines 60-61 of the revised manuscript).

Comments 3: The phrase “Most frequent food” (line 68) would probably be better to say “Most frequent foodstuff” by guessing its meaning of sentence.

Response 3: Agree. We have accordingly changed to “Most frequent foodstuff” (line 68 of the revised manuscript).

Comments 4: The meaning of the phrase “the optimal cut-offs and reference range that best discriminated” (lines 89-90) seemed easily and clearly understandable, but it seemed a little unnatural and weird, especially its wording. This phrase would probably be better to revise like “the optimal threshold and reference range that was the most suitable to discriminate actual patients from controls”.

Response 4: We have revised the sentence to “ We assessed the optimal threshold and reference range that was the most suitable to discriminate actual patients from controls” (lines 101-102 of the revised manuscript).

 “2. Materials and Methods”

 This section seemed to have no critical mistakes and defects, and the assay method was not new. So there was nothing to say specifically.

 “3. Results”

The obtained results were presented here in a good manner.

Comments 1: The phrase “in comparison to” (line 157) would probably say “in comparison with” or “as compared to”.

Response 1: We have corrected to “in comparison with ” (line 169 of the revised manuscript).

Comments 2: The word “probable” (line 178) was suspected to be “probability”.

Response 2: We appreciate the reviewer for pointing this out. However, we prefer to leave HIT probable as this is taken from the ELISA manufacturer recommendation as the reference ranges; as also described in the Materials and Methods section, 2.2. Determination of serum DAO concentration (lines 132-133 of the revised manuscript).

In order to avoid any confusion, we have added an explanation, and the sentence now reads, While the proposed reference for HIT probable (< 10 U/ml), suggesting that HIT is likely to be present, … (lines 190-191 of the revised manuscript).

 “4. Discussion”

Some parts of the Discussion seemed just repetition and the same to the description in other sections. To make the manuscript more concise and compact, it would probably be better, might be necessary to avoid such a repetition or try to rephrase those parts. The descriptive contents of this section seemed quite careful and well-considered. Particularly,

Comments 1: The phrase “most likely” (lines 220-221) might be better to say “the most likely condition” to make the meaning clear and easier to understand.

Response 1: We have corrected to “the most likely condition” (line 236 of the revised manuscript).

Comments 2: The phrase “evidence in favour of” (line 232) seemed a little unnatural and awkward, and it would probably be better to say simply “evidence for”.

Response 2: We have corrected to “evidence for” (line 247 of the revised manuscript).

Comments 3: The phrase “the diagnosis as HIT” (line 233) should be “the diagnosis of HIT”.

Response 3: We have corrected to “the diagnosis of HIT” (line 248 of the revised manuscript).

Comments 4  The phrase “the utility of measuring serum DAO activity in identifying” (line 255) seemed awkward, and it would probably be better to say “the usefulness of measuring serum DAO activity to identify”.

Response 4: We have rephrased to “the usefulness of measuring serum DAO activity to identify” (line 270 of the revised manuscript) as suggested by the Reviewer.

 “5. Conclusion”

The conclusion derived from the present study might be understandable and reasonable, and therefore publishable in this journal.

Thank you again very much for taking the time to review this manuscript and for constructive comments that improved our our manuscript.